# Evaluating large language model performance and reliability in scoring picture description tasks for neuropsychological assessment

Michael J. Kleiman[ID]*

Comprehensive Center for Brain Health, Department of Neurology, University of Miami Miller School of Medicine, Boca Raton, Florida, United States of America

* mjkleiman@miami.edu

## Abstract

Picture description tasks, such as the Cookie Theft task, are widely used in neuropsychological assessments to detect cognitive impairment. However, manual scoring is time-consuming, requires specialized training, and is subject to interrater variability. Recent advancements in natural language processing, particularly large language models (LLMs), offer a promising solution to automate and standardize the scoring process. This study evaluated the performance and reliability of five LLMs (GPT-4 Turbo, GPT-4o, Claude 3 Opus, Claude 3 Sonnet, and Llama 3 70b) in scoring the Cookie Theft picture description task. A subset of 25 participants were selected from the DementiaBank corpus. The LLMs were tasked with scoring 22 content units in the participants' responses using various prompt strategies, including few-shot learning, prompt chaining, and self-consistency. LLM performance was compared to the consensus score of three human raters. LLMs demonstrated comparable accuracy to human raters in scoring the Cookie Theft task, with no significant differences in mean absolute error (MAE) between the best performing models and human raters. Few-shot learning significantly improved LLM performance, while prompt chaining and self-consistency showed limited benefits. Claude 3 Opus and GPT-4o exhibited the highest accuracy and reliability. Notably, LLMs showed significantly higher interrater reliability compared to human raters. The findings demonstrate the potential of LLMs to accurately and reliably score picture description tasks, offering a promising approach to streamline and standardize neuropsychological assessments. By automating the scoring process, clinicians and researchers can benefit from increased efficiency, reduced subjectivity, and improved scalability in evaluating cognitive functions.

**Data availability statement:** Data was obtained from the DementiaBank corpus of TalkBank, and is made available upon request following free membership admission. See their website at dementia.talkbank.org for more information about registering for membership, obtaining access to data, and other rules. Ratings from models, human raters, and consensus are available via publicly available repository, doi:10.17605/OSF.IO/9FQNR.

**Funding:** This work was partly supported by funding provided by the Alzheimer's Association (AARF-22-923592) and the Evelyn F. McKnight Brain Research Foundation. The funders had no role in study design, data collection and analysis, decision to publish, or preparation of the manuscript.

**Competing interests:** I have read the journal's policy and the authors of this manuscript have the following competing interests: MJK is the founder of SciKey Diagnostics, a cognitive diagnostics company. He has received consulting fees from Hjarna Scientific, Wellsaid.AI, and Cognivue. He has received research grants from the Alzheimer's Association, American Academy of Neurology, Evelyn F. McKnight Brain Research Foundation, and the Florida Department of Health. No funding sources had any role in study design, data collection and analysis, decision to publish, or preparation of the manuscript.

## Author summary

Picture description tasks, where patients describe an image like the "Cookie Theft" picture, are valuable tools for detecting cognitive impairment and dementia. However, these assessments are infrequently used in large research studies with multiple sites because scoring can differ between raters, and it is time-consuming. In this study, we investigated whether artificial intelligence, specifically large language models (LLMs) like ChatGPT and Claude, could accurately score these assessments compared to humans. We compared the performance of five different LLMs to trained human raters when scoring picture descriptions from both cognitively normal and dementia participants. Our results show that LLMs can score these assessments with accuracy comparable to human raters, while also being more consistent across multiple evaluations of the same response. This finding is particularly important because it suggests that AI could help standardize the scoring process, making these valuable assessments more accessible in both clinical and research settings.

## 1. Introduction

Picture description tasks have been commonly used in neuropsychological assessments for over 50 years [1], with the Cookie Theft picture from the Boston Diagnostic Aphasia Examination (BDAE) being one of the most widely used stimuli. Originally developed to detect language deficits in aphasia, this task has since demonstrated utility in assessing a broader range of cognitive functions. It was later discovered that patients with early-stage Alzheimer's disease (AD) exhibited distinct patterns of performance on the Cookie Theft task, characterized by longer completion times and missing content units, suggesting its potential for detecting cognitive impairment in mild to moderate dementia [2,3]. Longitudinal studies have further established the relationship between changes in picture description performance and cognitive decline [4], observing that individuals with AD employed compensatory strategies in their descriptions as the disease progressed and highlighting the task's sensitivity to subtle linguistic changes over time. Advancements in natural language processing (NLP) techniques have enabled more detailed analyses of linguistic and lexical features in picture descriptions, revealing specific deficits in semantic knowledge, executive function, and working memory associated with cognitive impairment [5,6].

Despite the utility of picture description tasks in clinical cognitive assessment, they are not often utilized effectively in research settings. Accurate scoring requires specialized training, making assessment judgements comparatively time-consuming and labor-intensive to administer and interpret over quicker and simpler assessments such as the trail-making test [7]. Additionally, the reliance on subjective clinical judgment and lack of standardized scoring criteria have led to concerns about interrater reliability and limited scalability, especially when participants use non-standard dialects [8]. Furthermore, the need for expert raters has restricted accessibility in resource-constrained settings [9].

Recent advances in natural language processing have enabled automated analysis of picture descriptions, with studies utilizing various computational approaches [10]. Traditional NLP methods have focused on extracting linguistic features such as lexical diversity and syntactic complexity [11,12]. Machine learning approaches have demonstrated promise in automating scoring, though they often require extensive training data and may lack interpretability.

The rapid advancements in NLP and the emergence of large language models (LLMs) present a promising solution to these challenges. LLMs have demonstrated human-level performance on complex linguistic tasks, exhibiting the ability to capture subtle semantic and syntactic features relevant to cognitive assessment. The present study aims to investigate the feasibility and efficacy of tasking LLMs with scoring picture descriptions obtained from the DementiaBank corpus. We hypothesize that LLMs will produce accurate results comparable to human raters while exhibiting decreased variability in scoring. Utilizing LLMs to score assessments has the potential to enhance the efficiency, consistency, and accessibility of neuropsychological evaluation. Automating the scoring process would alleviate the need for extensive training and reduce the time and labor costs associated with manual scoring. Moreover, the standardization of scoring criteria through LLMs could improve reliability and facilitate the comparison of results across different studies and clinical settings.

## 2. Materials and methods

### 2.1. Data

The data used in this study was obtained from the DementiaBank [13] ADReSSo challenge [14], which consists of speech recordings of picture description tasks using the Cookie Theft image from the Boston Diagnostic Aphasia Examination [1]. Recordings were collected from both cognitively normal participants as well as cognitively impaired patients with a diagnosis of Alzheimer's disease. A subset of 25 participants, 13 classified as cognitively normal and 12 as cognitively impaired, were selected from the sample of 237 total participants with collected audio recordings of picture description tasks, stratified by cognitive status and matched where possible on age and gender. This sample size was determined by practical resources constraints: each recording required manual transcript verification, independent scoring by each of three human raters, and consensus discussion with a fourth neutral rater, totalling approximately 50 hours of human labor. This sample size provided adequate statistical power for within-subjects comparisons across the five LLMs and four prompt strategies (see 2.5.4 Statistical Power). Participant characteristics were controlled between groups where possible, given the limited demographics available; only age, sex, MMSE score, and a probable diagnosis were present in the dataset. The IDs and other characteristics of the selected participants can be found in **S1 Appendix**.

### 2.2. Content units

Content units (CUs) refer to the components of a picture description task that can be described by the participant and counted towards an accuracy rating of the participant's ability to describe the objects and actions occurring in the image. For example, describing a "boy" is one content unit, and the boy "standing on a stool" is another, however the participant stating that there is a "picture" in front of them does not describe the content contained in the image and thus is not considered a CU. Twenty-two CUs were chosen for the Cookie Theft image, derived from those identified in previous studies [15] but not including more subjective components such as the judgement of emotionality or intentionality, with the exception of overt displays (e.g., the girl's positive emotion).

Each content unit was also formed into a simple True/False question and used by both the human raters as well as the LLM models to provide scores for the presence or absence of each content unit. Examples were also generated based on custom written sample transcripts (not used in analysis), designed to elicit edge-cases and promote better understanding of what each content unit is measuring. Examples were provided to the human raters as well as the LLMs in the few-shot prompts. These questions and examples are provided in **S2 Appendix**.

### 2.3. Models

**2.3.1. Large language models.** Five LLMs were chosen for this study: OpenAI's GPT-4-Turbo ("gpt-4-turbo-0429") and GPT-4o ("gpt-4o-2024-05-13"), Meta's Llama 3 ("llama-3-70b-instruct"), and two of Anthropic's Claude 3 models: Sonnet ("claude-3-sonnet-20240229") and Opus ("claude-3-opus-20240229"). Other models were considered including Mixtral 8x7B, Mixtral 8x22B, GPT-Turbo-3.5, GPT-4, and Gemini 1.5 Pro, but they were unable to consistently output in JSON format, a necessary step to minimize manual data entry and ease preprocessing and analysis, and so were excluded. In compliance with the terms of TalkBank and DementiaBank, no data was uploaded using services that retain data for training; Llama 3 was ran on the University of Miami Pegasus supercomputer cluster, and both OpenAI and Anthropic specify in their terms that data uploaded via the API is not retained after processing.

**2.3.2. Prompt engineering.** Initial system prompts were generated to ensure that all models produced consistent responses to each question. Models were informed that they are "tasked with carefully examining a transcript of a picture description task and determining whether the participant has accurately described each component of the picture." They were then instructed "you will then consider the following list of questions, and carefully evaluate the transcript" before providing their answer. Following the questions, the assistant was reminded to output their answers as either True or False in a 1 or 0 format. All outputs were requested to be in JSON format, with examples provided.

Four different strategies were utilized to maximize accuracy and compared: chain-of-thought prompting, few-shot prompting, prompt chaining, and self-consistency. As the questions posed in this study are fairly straightforward True or False questions, more advanced techniques that may improve logical reasoning, such as tree of thought prompting [16], would likely not have been worth the increased processing costs involved in their implementation.

**2.3.2.1. Chain-of-thought prompting:** Chain-of-thought prompting works by guiding LLMs to first explain their reasoning, often in a stepwise format, before stating their answer [17]. When problems are complex, this strategy mimics that used in human problem-solving behavior by breaking down the problem into manageable sub-tasks. In practice, this works by causing the model to attend more to the user-provided input, in our case the participant's transcript, instead of relying on its training data which can cause incorrect judgements as well as hallucinations. This can result in the model's "attention" becoming more consistent even when questions are altered or presented in different ways [18]. For our use case, the models were asked to provide a one-to-two sentence explanation of their True/False answer prior to providing the answer itself.

**2.3.2.2. Few-shot prompting:** In few-shot prompting, models are provided examples of expected inputs and outputs, as opposed to "zero-shot" prompting where no explicit examples are provided and models are expected to use only their training data to guide their responses [19]. This method helps the model understand what is expected from its responses, including structure, format, and examples of reasoning. In addition, these examples provide a way for the prompt engineer to explain how to navigate potentially confusing scenarios by structuring the examples using expected outputs to given inputs. In our practice, we provide two sample transcripts that include example answers to often confusing or misunderstood questions. Few-shot prompting is only used alongside chain-of-thought prompting, as providing examples without the paired explanations would not be expected to improve accuracy.

**2.3.2.3. Prompt chaining:** Prompt chaining is an advanced technique that involves providing the model a sequence of prompts or questions to solve complex tasks, instead of combining all questions into a single prompt. This can allow the model to focus solely on each question, without basing their answer on previous responses, and in more complex questions can allow the model to iterate through multi-step reasoning leading to increased accuracy [20]. Conversely, prompt chaining adds complexity to prompt generation as well as increased computational overhead, and so chaining questions can be both beneficial and detrimental, depending on the particular task; if the task is complex, the increased complexity is outweighed by the benefits in improved reasoning, however simple tasks such as those examined in this study may not benefit enough or at all. Nonetheless, chaining was examined as a possible method of improving score accuracy by providing only a single question at a time, in addition to examples of only that question when few-shot prompting was utilized.

**2.3.2.4. Self-consistency:** Self-consistency is a prompt engineering technique aimed at improving the robustness and reliability of the outputs generated by LLMs through generating multiple independent responses and then selecting the most consistent or frequent answer among them [21]. This approach relies on the principle that the most accurate or plausible answers will naturally be more consistent across iterations, thereby reducing the influence of outliers and errors. While some approaches utilize slight variations in questions to elicit variation in responses, our approach was to collect three responses to the same prompt from each model and use the median answer for each question, and so the same questions and examples were provided for each run of the model.

## 2.4. Procedure

### 2.4.1. Transcript generation.
Speech recordings were first transcribed automatically using *WhisperX* [22], a wrapper of OpenAI's *Whisper* speech recognition model [23] that integrates voice activity detection and forced phoneme alignment to ensure word-level timings are as accurate as possible while also reducing hallucinations and error rates. The use of automatic speech recognition significantly reduces time and effort in transcription, and the *Whisper* offering produces significantly higher accuracy than many other free and paid options. Additionally, the model is fully open-source and capable of being run on in-house hardware. The *Large-v3* model offered by OpenAI was used. In most recordings, the voice of the experimenter or clinician was present. The diarization function of the *WhisperX* model was utilized to identify portions of recordings where different speakers were present.

Following the creation of the automated transcriptions, manual correction was performed on each to ensure that only the participant's response was present in the transcript, with all speech from the clinician or experimenter removed including instruction portions as well as interjections including "do you see anything else?". While the generated transcriptions were typically accurate, occasional hallucinations were present such as repeated words or phrases ("the boy the boy the boy the boy") or finding non-existent speech in background chatter. When present, the audio recording was ran through the *WhisperX* model a second time, which often fixed the errors while maintaining accurate word-level timings. When this did not solve the problem, transcripts and timings were manually corrected by the PI (MJK).

### 2.4.2. Human scoring and consensus.
Three raters were tasked with determining whether each participant's picture description transcript contained the twenty-two CUs, using the provided questions in **S2 Appendix**, assigning each a score of True or False. The raters consisted of the PI (MJK) as well as one trained neuropsychologist and one layperson, each of whom were trained on how to properly score picture description tasks as well as the background of the study. The layperson with additional training was chosen to increase real-world validity by approximating the performance of a newly trained neuropsychologist or one who does not frequently score these tasks. After scoring, the three raters along with an additional trained neuropsychologist met to determine consensus for each CU. Any disagreement led to continued discussion until agreement was reached amongst all parties. The consensus scoring was then determined as "ground truth", permitting scoring of each human rater as well as the scores determined by each LLM.

### 2.4.3. LLM scoring.
Each LLM was provided the twenty-two questions as well as additional prompting based on the prompt engineering strategy (chain-of-thought, zero-shot vs few-shot, prompt chaining). Responses were formatted as JSON objects and transformed into *pandas* dataframes to facilitate comparative analysis.

## 2.5. Analysis

### 2.5.1. Examination of prompt strategies.
Each model used in this study was provided four separate prompts for each transcript; chain-of-thought without examples ("Zero-shot"), chain-of-thought with examples ("Few-shot"), few-shot with questions separated via chaining ("Chaining"), and self-consistency using three independent Few-shot runs ("Self-Consistency"). Prompts without using chain-of-thought ("Basic"), where only the list of questions were provided in addition to limited instructions, did not produce valid JSON code in all models and so were excluded.

**2.5.2. Examination of ratings.** The outputs of each model's prompt strategy as well as each individual human rater were compared to the "ground truth" consensus ratings for each of the 22 CUs, producing a mean absolute error ("MAE") score through the count of each rater's deviation from ground truth. The MAE was used as the dependent variable when comparing models and raters to each other, using repeated measures Analyses of Variance (ANOVAs) and pairwise t-tests with Bonferroni correction within the *pingouin 0.5.4* Python package [24]. Interrater reliability for both humans and LLMs (using the three self-consistency runs) were examined using *pingouin*'s intra-class correlation function which generates ICC coefficients for each class (human or LLM) based on the three human raters or the three self-consistency results. The ICC2k "average of random raters" method was used, which estimates reliability for the average of 3 raters that would be reasonably expected to generalize to a larger population of raters given the heterogeneity of rater experience with neuropsychological assessment examination for the human raters as well as the stochastic nature of machine learning models for the LLM raters.

**2.5.3. Human/machine comparisons.** The MAE of both humans and LLMs were compared using t-tests comparing the median performance of the two groups, and repeated measures ANOVAs with participants used for within-subjects comparison and examined using pairwise t-tests to measure changes when considering all raters and model runs as well as when examining individual participant transcripts. T-tests were also used to compare the differences between humans and machines with respect to each of the twenty-two CUs. Statistical analyses were performed using the *pingouin 0.5.4* Python package.

**2.5.4. Statistical power.** Power analyses were conducted using G*Power 3.1.9.7 for the primary analysis comparing LLM and human rater performance. For repeated measures ANOVAs comparing prompt strategies (4 levels) and model types (5 levels) with $\alpha = 0.05$ and $n = 25$, and assuming moderate correlations among repeated measures ($r = 0.5$), the study achieved 80% power to detect effect sizes of $d = 0.3$-$0.4$ (small-to-medium effects). For human vs LLM comparisons using two-level repeated measures ANOVAs, 80% power was achieved for $d = 0.6$, (medium effect).

## 3. Results

### 3.1. Participant characteristics

Very few measures were available for participants within the DementiaBank dataset, including age, gender, and the Mini-Mental State Exam (MMSE) score which was used to determine dementia classification. Age and gender were compared between participants with normal cognition and probable AD, revealing no significant differences (**Table 1**). MMSE scores were significantly different, $t(12)=5.79$, $p < .001$, which is unsurprising as this measure was used to determine the classification of probable AD.

### 3.2. Human rater comparisons

When calculating MAE compared to the consensus "ground truth" scores, the three human raters (AS: $1.12 \pm 0.88$ MAE; KR: $1.28 \pm 1.21$ MAE; MK: $1.20 \pm 1.44$ MAE) did not significantly differ from each other as determined by a one-way repeated measures ANOVA, $F(2,72)=0.11$, $p = .895$. The mean of the three raters' MAE scores across all 25 transcripts

**Table 1. Participant characteristics.**

|  | Controls | Probable AD | p-val |
|---|---|---|---|
| **age** | 67.31 (6.18) | 66.92 (8.70) | 0.897[t] |
| **gender** | 0.62 (0.51) | 0.58 (0.51) | 0.806[x] |
| **mmse** | 28.69 (0.95) | 19.42 (5.47) | **0.000[t]** |

t = t-test; χ = chi-squared test; bold = significance.

was 1.20 ± 1.19. When examining performance on individual questions, the sum total MAE across all participants only exceeded a total score of five and mean of 2.0 on three questions: Question 3 ("action of reaching for cookies") (M = 5.7, SD = 4.2, Σ = 17), Question 7 ("action of the stool tipping over") (M = 4.0, SD = 2.6, Σ = 12), and Question 10 ("action of a girl taking or being given cookies") (M = 3.0, SD = 2.0, Σ = 9).

The intraclass correlation coefficient (ICC) was calculated to assess interrater reliability among the three human raters. The results showed an ICC of 0.347 (95% CI [-0.30, 0.69], F(24, 48) = 1.51, $p = .110$, $\eta^2 = 0.431$), indicating poor reliability and a high degree of inconsistency among the raters, **Table 2**.

### 3.3. LLM comparisons

**3.3.1. Prompt strategy.** A one-way repeated measures ANOVA examining each of the four prompt strategies (zero-shot chain-of-thought, few-shot chain-of-thought, prompt chaining, and self-consistency) revealed significant effects of prompt strategy, F(3,72) = 8.55, $p < .001$, $d = 1.19$. Post hoc pairwise comparisons with Bonferroni correction were performed, revealing that the Zero-Shot (M = 1.58, SD = 1.25) and Chaining (M = 1.72, SD = 1.37) strategies were not significantly different from each other (p = .315) but performed significantly worse than the other two models, Few-Shot (M = 1.25, SD = 1.33) and Self-Consistency (M = 1.19, SD = 1.29), in all comparisons (Zero-Shot vs Few-Shot, $p = .007$; Zero-Shot vs Self-Consistency, $p = .002$; Chaining vs Few-Shot, $p = .004$; Chaining vs Self-Consistency, $p = .002$). Few-Shot and Self-Consistency were also not significantly different from each other, p = .356, suggesting that across all models a single Few-Shot run is comparable to averaging the results of three runs and thus the single run is preferable to maintain accuracy while reducing compute costs.

**3.3.2. Model type.** A one-way repeated measures ANOVA examining each of the five models (GPT4 Turbo, GPT 4o, Claude 3 Opus, Claude 3 Sonnet, Llama 3 70b) identified a significant main effect of model type, F(4,96) = 3.20, $p = .016$, $d = 0.73$, **Fig 1**. Further exploration using post hoc pairwise t-test comparisons with Bonferroni correction for multiple comparisons revealed that the only significant difference between models was between the Claude 3 Sonnet (M = 1.67, SD = 0.83) and GPT4o (M = 1.24, SD = 1.04) models, $p = .026$.

**3.3.3. Interaction between model and prompt strategy.** A two-way repeated measures ANOVA with model type and prompt strategy as within-subjects factors was ran to examine the interaction between model type and prompt strategy, and determine the best possible combination or combinations of factors. The ANOVA was significant, F(12,288) = 2.78, $p = .001$, $d = 0.31$. Post hoc analysis using pairwise t-test comparisons with Bonferroni correction for multiple comparisons revealed that this interaction was localized to the difference between Claude 3 Sonnet (M = 2.48, SD = 1.12) and GPT 4o (M = 1.24, SD = 1.30) using the Chaining prompt strategy, $p = .001$, $d = 1.02$ (**Fig 1**), as well as within both Claude models and the differences between the Chaining prompt strategy, which always performed the worst, and the Few-Shot and Self-Consistency strategies which performed the best. No other comparisons within models were significant.

**Table 2. Comparison of human and machine reliability across 3 ratings.**

|  | ICC | F | df1 | df2 | $\eta^2$ | pval |
|---|---|---|---|---|---|---|
| **Humans** | 0.347 | 1.514 | 24 | 48 | 0.431 | 0.110 |
| **GPT-4 Turbo** | 0.889 | 8.836 | 24 | 48 | 0.815 | **<0.001** |
| **GPT-4o** | 0.941 | 16.229 | 24 | 48 | 0.890 | **<0.001** |
| **Llama 3** | 0.961 | 26.391 | 24 | 48 | 0.930 | **<0.001** |
| **Claude 3 Opus** | 0.934 | 14.889 | 24 | 48 | 0.881 | **<0.001** |
| **Claude 3 Sonnet** | 0.925 | 13.177 | 24 | 48 | 0.868 | **<0.001** |

Bold = significance.

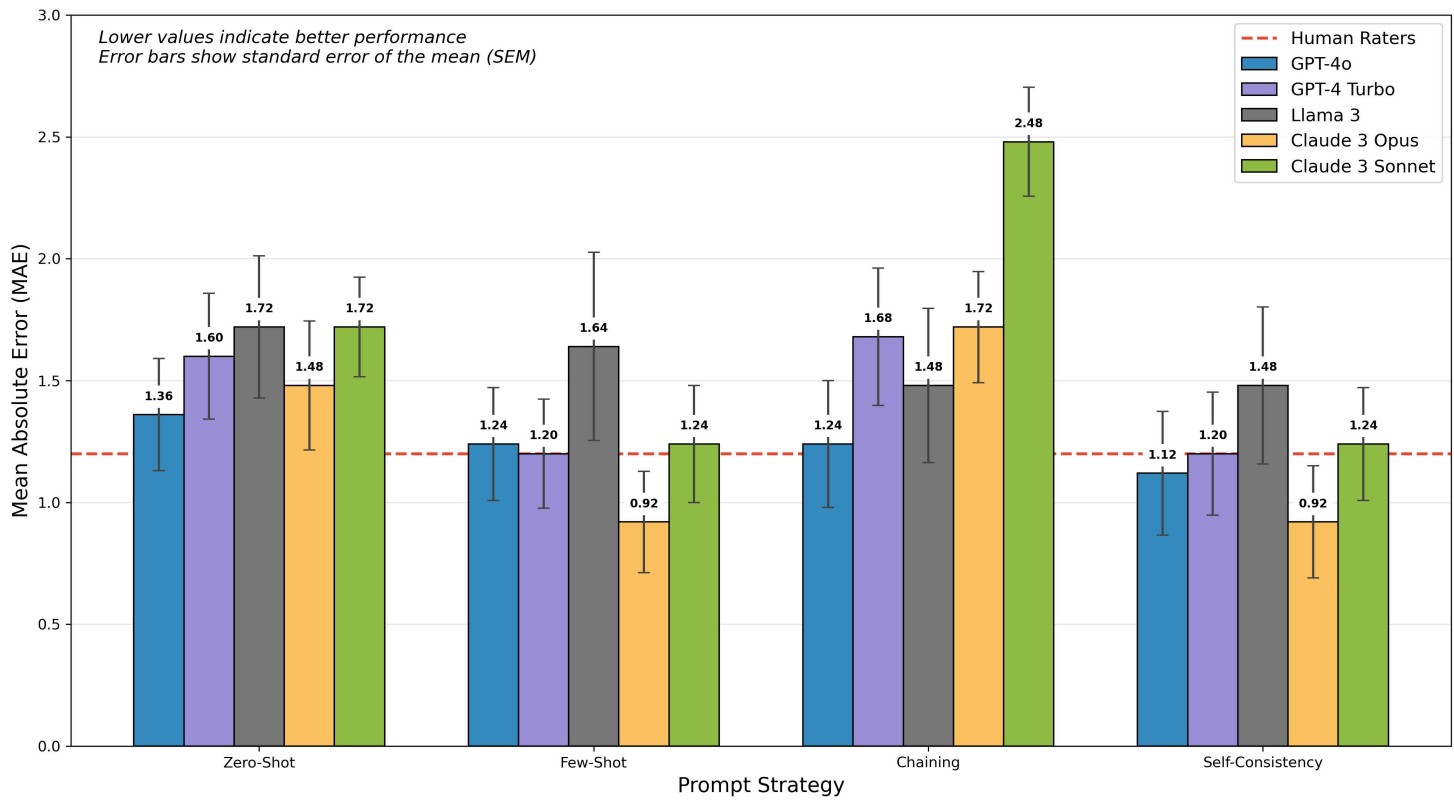

**Fig 1. Effect of prompt strategies on model performance.** Mean absolute error (MAE) is shown for each model and prompt strategy, with lower values indicating better performance. Error bars represent the standard error of the mean (SEM). Few-shot and self-consistency strategies outperformed other approaches, with the best performing models (GPT-4o and Claude 3 Opus) achieving accuracy comparable to human raters when using few-shot or self-consistency prompting. No difference was found between these two prompting strategies, suggesting that the self-consistency strategy is unnecessary.

As the single-run Few-Shot model was determined to be the most efficient and effective prompt strategy, the three Few-Shot runs used to calculate the median Self-Consistency score were included in a repeated measures ANOVA model to determine whether any one model's output was more or less variable than other models. No significant differences were found for any one model across all three runs. ICC2k calculations also revealed excellent reliability for every model (**Table 2**). Repeated measures ANOVAs with model type and model run as within-subjects factors revealed no significant differences between model type, $F(4,96)=2.18$, $p=.011$, $d=0.60$ (**Fig 1**).

### 3.4. Differences between humans and LLMs

To determine the scoring ability of each model and compare with human raters, repeated measures ANOVAs comparing MAE scores for the three humans and each model's three Few-Shot runs. No significant differences between human MAE and any one model's MAE were observed: GPT-4 Turbo, $F(1,24)=1.06$, $p=.313$; GPT-4o, $F(1,24)=.01$, $p=.947$; Llama 3, $F(1,24)=2.17$, $p=.153$; Claude Sonnet, $F(1,24)=0.22$, $p=.643$; and Claude Opus, $F(1,24)=1.55$, $p=.226$. Item-level analysis can be found in **Table 3**.

**Table 3. Mean absolute error totals for each question, averaged by rater.**

|  | Humans | GPT-4 Turbo | GPT-4o | Llama 3 | Claude 3 Opus | Claude 3 Sonnet |
|---|---|---|---|---|---|---|
| **Question 1** | 0.33 (0.58) | 1.0 (0.0) | 1.33 (0.58) | 1.0 (0.0) | 1.0 (0.0) | 1.0 (0.0) |
| **Question 2** | 0.67 (0.58) | 1.33 (0.58) | 1.0 (0.0) | 1.0 (0.0) | 1.33 (0.58) | 3.33 (1.53) |
| **Question 3** | 5.67 (4.16) | 2.33 (0.58) | 2.0 (0.0) | 3.67 (0.58) | 2.33 (0.58) | 3.67 (0.58) |
| **Question 4** | 1.67 (1.15) | 3.0 (0.0) | 1.67 (0.58) | 2.0 (0.0) | 2.33 (0.58) | 1.0 (0.0) |
| **Question 5** | 0.67 (0.58) | 0.67 (0.58) | 0.67 (0.58) | 0.33 (0.58) | 0.0 (0.0) | 0.33 (0.58) |
| **Question 6** | 1.33 (0.58) | 0.0 (0.0) | 0.33 (0.58) | 1.33 (1.53) | 3.0 (0.0) | 2.67 (1.15) |
| **Question 7** | 4.0 (2.65) | 3.0 (1.0) | 3.33 (0.58) | 6.0 (0.0) | 2.33 (0.58) | * 1.0 (0.0) |
| **Question 8** | 1.67 (0.58) | 1.67 (0.58) | 2.0 (0.0) | 1.33 (0.58) | * 0.0 (0.0) | 0.33 (0.58) |
| **Question 9** | 0.67 (0.58) | 1.0 (0.0) | 1.0 (0.0) | 1.33 (0.58) | 1.0 (0.0) | 1.0 (0.0) |
| **Question 10** | 3.0 (2.0) | 4.33 (0.58) | 4.0 (1.73) | 4.67 (1.53) | 2.67 (0.58) | 3.67 (0.58) |
| **Question 11** | 0.33 (0.58) | 3.33 (0.58) | 0.33 (0.58) | 1.0 (1.0) | 0.0 (0.0) | 0.0 (0.0) |
| **Question 12** | 1.0 (1.73) | 0.0 (0.0) | 1.0 (0.0) | 1.0 (0.0) | 0.0 (0.0) | 0.0 (0.0) |
| **Question 13** | 1.33 (0.58) | 1.0 (0.0) | 1.0 (0.0) | 1.33 (0.58) | 1.0 (0.0) | 1.33 (0.58) |
| **Question 14** | 0.33 (0.58) | 0.0 (0.0) | 0.0 (0.0) | 0.0 (0.0) | 0.0 (0.0) | 0.0 (0.0) |
| **Question 15** | 1.0 (1.0) | 3.0 (1.0) | 0.33 (0.58) | 2.0 (0.0) | 0.33 (0.58) | 0.67 (0.58) |
| **Question 16** | 1.67 (0.58) | 2.67 (1.15) | 2.0 (0.0) | * 5.0 (0.0) | 1.67 (0.58) | 1.0 (0.0) |
| **Question 17** | 0.67 (0.58) | 1.0 (0.0) | 1.33 (0.58) | 3.0 (0.0) | 1.33 (0.58) | 4.33 (0.58) |
| **Question 18** | 1.33 (0.58) | 3.0 (0.0) | 3.0 (0.0) | 1.67 (0.58) | 2.67 (0.58) | 5.0 (0.0) |
| **Question 19** | 0.33 (0.58) | 1.0 (1.0) | 1.0 (0.0) | 0.0 (0.0) | 0.33 (0.58) | 0.0 (0.0) |
| **Question 20** | 0.0 (0.0) | 0.33 (0.58) | 0.0 (0.0) | 0.0 (0.0) | 0.0 (0.0) | 0.0 (0.0) |
| **Question 21** | 1.33 (0.58) | 1.0 (0.0) | 1.0 (0.0) | 1.0 (0.0) | 1.0 (0.0) | 1.0 (0.0) |
| **Question 22** | 1.0 (1.0) | 0.67 (1.15) | 1.33 (0.58) | 0.33 (0.58) | 0.0 (0.0) | 1.0 (1.0) |

Scores calculated using sum of all 25 participants with Mean (SD) of raters/runs for each question, separated by model. Statistical comparisons performed using repeated measures ANOVAs.

\* = the model significantly differs from human raters.

## 3.5. Question comparisons

Repeated measures ANOVAs were ran comparing the three Few-Shot runs with the three human rater scores for each of the twenty-two questions, to determine whether any model performed significantly better or worse than the human raters. From the 110 ANOVAs ran, only three comparisons were significant pre-correction: Claude 3 Sonnet for Question 7 (stool tipping over), $F(1,24)=6.61$, $p=.017$, $d=1.05$; Claude 3 Opus for Question 8 (falling off a stool), $F(1,24)=6.00$, $p=.022$, $d=1.00$; and Llama 3 for Question 16 (overflowing sink), $F(1,24)=4.36$, $p=.047$, $d=0.85$. Llama 3 for Question 16 performed worse than the human raters, while both Claude 3 models performed better (**Table 3**). Question 7, involving the stool tipping over, was occasionally scored as present in error when only the action of the boy falling off the stool is described. For example, subject 229's statement *"I thought the kid's going to fall off the stool"* was commonly misinterpreted by both human raters and most models as describing the stool about the fall, instead of the boy. The opposite was observed for question 8, which describes the action of the boy falling off the stool; while occurring less often than question 7, both human raters and models occasionally correctly marked present the boy standing on the stool (question 6) and the stool being described as tipping over (question 7), followed by scoring question 8 as true even when the action of falling off was not described. In subject 286's transcript, they describe *"this boy's on the stool getting cookies. His stool's about to fall"* which does not describe the action of the boy himself falling. Despite this, it was incorrectly rated as present by 1/3 human raters and GPT-4o. Finally, question 16 ("did the participant describe a sink that is overflowing?") was often scored

as present when only water was described as being on the floor (question 17) rather than the sink being the thing that is overflowing; for example, subject 218 describes *"the water's spilling on the floor"* which was scored incorrectly by 1/3 human raters and at least one run of every model.

## 4. Discussion

The key findings of this study highlight the effectiveness of LLMs in scoring neuropsychological assessments, achieving performance comparable to human raters while offering notable advantages in consistency and speed. Specifically, while error rates between human and machine scoring were statistically similar in terms of accuracy, LLMs showed superior consistency across multiple scoring runs indicated by ICC scores compared to human raters, whose variability was comparatively high. These results suggest that LLMs could reduce the subjective variability inherent in human scoring, thereby enhancing reliability and reproducibility in clinical and research applications.

The use of language models for interpreting clinical data is exciting, but controversial. Some researchers argue that language comprehension can be highly inconsistent and result in missed cues, leading to potentially erroneous conclusions, however our results do not agree with these conclusions. This may be due to differences in methodology and/or advancements in LLM technology since publication. For example, while we observed that chain-of-thought and few-shot prompting significantly increased model accuracy in our results, a study by Dentella et al. [25] did not incorporate these strategies as explored in our study, leading them to conclude that humans performed better overall than LLMs. Further, their most advanced model used was GPT-4, which was found to produce inconsistent results in preliminary testing resulting in its exclusion from this study. Chamieh et al. [26] explores answer scoring with similar short-form questions as in this study, comparing few-shot and zero-shot prompting strategies. Their findings suggest that LLMs are unable to accurately score these answers, but utilized outdated models and no chain-of-thought reasoning which may have impacted performance.

No significant differences were found between the Few-Shot and Self-Consistency prompting strategies, suggesting that a single Few-Shot run is sufficient for most use cases. Although Self-Consistency slightly improved error rates using the median outputs of three runs, the marginal gains in accuracy likely do not justify the additional computational costs in routine applications. Among the models tested, Claude Opus consistently performed the best and demonstrated the highest reliability, albeit at a substantially higher operational cost compared to GPT-4o, the second-best performer.

### 4.1. Model-specific performance

Analysis of model-specific performance revealed distinct strengths and trade-offs among the tested LLMs, emphasizing the importance of balancing consistency, accuracy, and cost in model selection. Anthropic's Claude Opus consistently delivered superior performance, outperforming other models in terms of error rates and variability. However, this came at a significantly higher computational cost, limiting scalability and utility in cost-sensitive applications. In contrast, OpenAI's GPT4o demonstrated similarly robust reliability and minimal variability between runs. Despite slightly lower overall accuracy compared to Opus, GPT4o was significantly more cost efficient: at the time of writing, Opus cost $15 per million input tokens and $75 per million output tokens, compared to GPT4o's lower rates of $2.50 per million input tokens and $10 per million output tokens, a reduction by a factor of six or more. To enable cost comparison, we calculated the average transcript length used in this study as 147.5 token, with the system prompt for few-shot prompting at 2934 tokens (3081.5 input tokens), with the average output including explanations and scoring producing 504.4 output tokens. Using these averages, Opus costs approximately $0.0841 per transcript, in comparison to GPT4o's $0.0127 per transcript.

Conversely, other models such as Claude Sonnet and Llama 3, while performing adequately, displayed greater variability and lower consistency which may limit utility in neuropsychological assessment scoring. While Sonnet was similar in price to GPT4o, it did not match its performance or consistency; while it benefitted from the robust design of the Claude architecture in its overall accuracy, it clearly lacked optimization for precision-intensive tasks as explored in the

assessment scoring task. Llama 3 presented the most variability among the tested models, with lower reliability and accuracy compared to both Claude models and GPT4 variants, potentially limiting its applicability in assessment scoring. However, Llama 3's open-source nature and its ability to be easily modified and fine-tuned may appeal to research or institutional settings that prioritize affordability over peak performance, especially as fine-tuning on similar assessments may improve overall performance.

## 4.2. Prompt engineering

Neuropsychological assessment scoring demands high levels of accuracy and consistency, necessitating a thorough evaluation of prompt engineering strategies to identify approaches that best align with these objectives. Further, it is crucial to consider the cost implications associated with LLMs, and they can be expensive to run and may require a delay in processing time if batching is performed to reduce overhead costs. Therefore, a comprehensive exploration of multiple prompt engineering techniques is warranted to determine the optimal parameters for answer scoring tasks, balancing performance and efficiency.

In addition, it is important to consider the differences between LLMs in the context of assessment scoring. While all models tested demonstrated adequate performance, notable variations were observed in terms of accuracy, consistency, and computational cost. Understanding these model-dependent effects is essential for selecting the most appropriate LLM for a given application, considering factors such as the required level of precision, available resources, and scalability.

It is important to note that the relatively straightforward nature of the answer scoring task in this study may not necessitate the use of more advanced prompt engineering strategies. The True/False questions posted to the LLMs do not require complex reasoning or multi-step problem-solving, which are the primary scenarios in which sophisticated methods such as tree of thought prompting [16] or recursive task decomposition [27] have been shown to be beneficial. Consequently, the additional computational overhead associated with these techniques may not be justified for the current task, as the marginal gains in accuracy are likely to be minimal.

**4.2.1. Chain-of-thought prompting.** The incorporation of chain-of-thought prompting, which allows the model to verbalize its reasoning process, has been shown to enhance the performance of language models in various tasks [17,28]. This technique emulates the "thinking out loud" strategy employed by newer reasoning models, such as GPT-o1, which has demonstrated improved accuracy and consistency in problem-solving and decision-making tasks. Interestingly, in our study, the absence of chain-of-thought prompting led to such poor performance in many models that they failed to generate a valid syntax, and when manually corrected in preliminary testing was found to produce error rates significantly higher than all other models. The incorporation of chain-of-thought, either through explicit prompting or through the use of reasoning models, is clearly critical to retrieving accurate responses from LLMs.

**4.2.2. Few-shot prompting vs self-consistency.** The use of few-shot prompting and self-consistency prompting strategies has garnered significant attention in recent studies, demonstrating their potential to enhance performance and reliability. Chamieh et al. [26] described the effectiveness of few-shot prompting in answer scoring tasks, albeit without utilizing chain-of-thought reasoning. Our findings support this observation, with few-shot significantly outperforming zero-shot strategies when examining error rates. In contrast, while Wang et al. [21] showcased improved accuracy and robustness of the self-consistency technique in complex question-answering scenarios compared to few-shot alone, our findings observed no significant difference between a single few-shot run and the median or mean scores of multiple. However, it is important to note that our study did not fully utilize the "multiple reasoning paths" principle of self-consistency highlighted in that study, as the nature of answer scoring is relatively simplistic and does not require advanced reasoning for each question. This limitation may explain the discrepancy between our findings and those observed by Wang et al.

Interestingly, our analysis revealed that the GPT-4o run used to calculate the Few-Shot measure performed significantly worse than the three other Few-Shot runs used to calculate the Self-Consistency metric, although no significant

difference was found across all participants. This observation suggests that the effectiveness of few-shot prompting may be sensitive to the specific examples provided and the model's ability to generalize from those examples. In contrast, the self-consistency approach, which relies on the model's ability to generate multiple consistent reasoning paths, appears to be more robust to variations in the input prompts.

### 4.2.3. Challenges of zero-shot prompting.

The absence of examples provided in zero-shot prompting can lead to inconsistent and unreliable responses from LLMs when answering ambiguous questions, as the models lack explicit guidance on the expected format and content of their outputs. In contrast, providing examples through few-shot prompting allows the model to answer based on expected norms and ensures consistency between participants [19], especially when questions are difficult to parse or poorly worded. The superiority of few-shot prompting over zero-shot approaches in this specific domain highlights the importance of carefully designing prompts that include representative examples to ensure optimal LLM performance in clinical and research applications.

### 4.2.4. Limitations of chaining.

The use of prompt chaining, where the entire prompt chain including instructions, the participant's response that is being scored, and all few-shot examples is provided for each question can lead to significant limitations in terms of processing time and computational costs. As each chained prompt requires the inclusion of the complete set of instructions and examples, the number of input tokens increases substantially for each participant, resulting in longer processing times and higher expenses associated with LLM usage. This is particularly relevant when considering the pricing structure of LLMs, which often charge per token or per query.

In our study, we observed that the chaining approach consistently performed worse than the few-shot and self-consistency strategies across all tested models (**Fig 1**). This suboptimal performance may be attributed to the increased complexity and token count of the chained prompts, which may introduce additional noise or ambiguity that hinders the model's ability to effectively attend to relevant inputs [29]. The current architecture of transformer-based language models relies heavily on the attention mechanism, which may struggle to effectively process and integrate information across longer sequences. In the context of prompt chaining, where each question is presented along with the entire instruction and few-shot prompt, the model is tasked with attending to a substantial amount of information per question, potentially leading to a degradation in its ability to focus on the most relevant aspects of the input. Overall, given the computational overhead associated with processing these longer prompts can lead to practical challenges in terms of scalability and resource allocation, particularly in resource-constrained settings or when dealing with large participant populations, the reduced or negligible performance of this strategy is not recommended for simple answer scoring tasks.

## 4.3. Application of findings

The findings from this study have several potential applications for enhancing the efficiency and consistency of neuropsychological assessment scoring in both clinical and research contexts. First, the use of automated speech recognition systems such as WhisperX [22] can significantly reduce the time and effort required for transcription while maintaining high accuracy. Recent research has validated the readiness of speech-to-text technology for analyzing free-spoken responses, even in diverse populations [30]. By integrating these automated transcription methods with LLM-based scoring, the entire workflow from assessment administration to score generation can be streamlined, leading to reduced costs and increased accessibility for clinics and research institutions.

Furthermore, the superior consistency demonstrated by LLMs in scoring picture description tasks compared to human raters highlights their potential to enhance the reliability of neuropsychological assessment data. Interrater reliability is a significant challenge in multi-site or multi-investigator studies, where variability in scoring practices can introduce noise and limit the interpretability of results. By employing automated methods, such as LLMs trained on standardized prompts and examples, large-scale studies can ensure greater consistency in scoring across different sites and raters, improving the quality and generalizability of the collected data.

The cost-efficiency of LLM-based scoring also has important implications for expanding access to neuropsychological assessments in low-resource settings. Clinical practices and research institutions in underserved areas often face financial and logistical barriers to the implementation of comprehensive assessment protocols. The use of open-source models like Llama 3 or DeepSeek R1, combined with the ability to fine-tune these models on specific tasks, offers a promising avenue for developing affordable and scalable solutions for assessment scoring in these contexts. Additionally, the effectiveness of few-shot prompting and the minimal computational overhead associated with single-run strategies suggest that even resource-constrained settings can benefit from the improved accuracy and consistency offered by LLMs without incurring substantial costs. As LLM technology continues to advance, further exploration of alternative scoring methods for existing assessments and the development of novel assessment paradigms tailored to LLM strengths may open new frontiers in neuropsychological research and clinical practice.

### 4.4. Limitations

One limitation of the study was the need to finalize the questions used for scoring prior to running all models, in order to ensure that human raters could utilize the same questions to support their answers. Consequently, the questions could not be modified in response to commonly missed points by human raters or iterated upon beyond the initial stages of the study. Future research should investigate strategies for improving answer scoring prompts, potentially through an iterative process that allows for refinement based on human and model performance.

Another limitation was the varying levels of expertise among the human raters in interpreting neuropsychological assessments. While no significant differences were found between raters, only two of the three raters had prior formal training in rating neuropsychological or cognitive assessments. The third rater received training in rating the picture description Cookie Theft task, but lacked broader experience in the field. This variability in rater background may have introduced some inconsistencies in the human scoring process, although the lack of significant differences suggests that the training provided was sufficient for the purposes of this study. An important limitation is the lack of comprehensive demographic data in the DementiaBank corpus. Education level, linguistic background, and socioeconomic factors were not available, limiting our ability to assess how these variables might influence LLM scoring accuracy. Prior research has demonstrated that dialectical variation can affect picture description scoring systems [8], and educational attainment influences linguistic complexity in cognitive assessments. Future studies should examine LLM performance across diverse demographic groups.

Finally, this study was unable to incorporate newly developed reasoning models, such as GPT-o1 and DeepSeek R1, which show promise for enhancing the accuracy and consistency of LLM-based scoring. At the time of initial analysis, these models did not exist, and during writing these models were rate-limited to a prohibitive extent and did not support structured outputs in JSON format, precluding their inclusion in the current study. However, as these limitations have been addressed by the time of publication, future research is planned to explore the performance of reasoning models on an expanded dataset, potentially offering further insights into optimizing LLM-based neuropsychological assessment scoring.

### 4.5. Ethical considerations

The deployment of LLMs for neuropsychological assessment scoring raises important ethical considerations that warrant careful examination before clinical adoption. While the findings of this study demonstrate technical feasibility and comparable accuracy to human raters, there are inherent biases, privacy issues, and interpretability constraints that must be addressed before these findings are implemented in clinical practice.

Modern LLMs are trained predominantly on standard English corpora, potentially introducing systemic biases in interpreting non-standard dialects, regional language variations, and speech patterns from diverse cultural backgrounds. Cognitive assessment tools have historically exhibited cultural biases leading to misdiagnosis or inappropriate decisions. For example, when scoring picture descriptions LLMs may misinterpret uncommon language correctly describing an object

or action as incorrect. Future validation must explicitly examine LLM accuracy across stratified demographic samples, including speakers of various dialects, multilingual individuals, persons with varying educational backgrounds, and participants from diverse socioeconomic contexts. Additionally, data privacy concerns arise when processing sensitive health information through commercial APIs. While major providers offer enterprise agreements specifying that API-submitted data is not retained for training, researchers and clinicians must verify compliance with HIPAA and applicable regulations. Open-source models deployed on-premises offer greater control but require substantial resources.

The opacity of LLM decision-making presents challenges for clinical interpretability and accountability. Unlike human raters who articulate reasoning through calibration discussions, neural networks generate predictions through complex transformations of multi-dimensional data, preventing straightforward explanation of decisions. Chain-of-thought prompting partially addresses this by requiring models to explain their reasoning, however these explanations represent post-hoc rationalizations rather than direct insights into internal computations and may not fully capture decision-driving features. This lack of interpretability has important implications: when automated scoring produces unexpected results, clinicians cannot interrogate model reasoning as they would with human raters. Furthermore, unlike human raters, models cannot be easily course-corrected; simply telling a model that they are incorrect does not cause it to respond differently in the future. Instead, the initial prompts must be carefully modified to ensure instructions are clear and interpretable.

When LLMs fail to follow instructions, or produce incorrect information, these are often referred to as "hallucinations"; however, it is more accurate to state that everything produced by a LLM is a hallucination, given that it is a simply a word-prediction model with a massive memory that is incapable of distinguishing between correct and incorrect information. Nonetheless, careful prompting can result in a consistent output as demonstrated by this study, and future implementations of LLMs (including models that have been developed following the drafting and analysis of this paper) are rapidly improving accuracy and decreasing hallucination rates. Of course, the rapid pace of LLM development introduces additional challenges. Commercial providers regularly update systems, potentially altering performance despite identical prompts, and typically deprecate older versions within 6–12 months; establishing quality control procedures can identify performance changes and maintain reliability.

## 5. Conclusion

This study demonstrates the potential of LLMs to accurately score picture description tasks, such as the Cookie Theft task, with performance comparable to human raters. Across various state-of-the-art LLMs and prompt engineering strategies, the study found that LLMs consistently produced accurate results while exhibiting decreased variability in scoring. The incorporation of chain-of-thought prompting and few-shot examples significantly improved LLM performance, with models like Anthropic's Claude Opus and OpenAI's GPT4o demonstrating the highest accuracy and reliability. These findings suggest that LLMs can be effectively leveraged to streamline and standardize the assessment scoring process. By automating the scoring of picture description tasks, clinicians and researchers can benefit from increased efficiency, reduced subjectivity, and improved scalability in the evaluation of cognitive function. This study lays the groundwork for the development of AI-assisted tools that can complement traditional assessment methods.

## Supporting information

**S1 Appendix. Sample characteristics of selected participants.**
(PDF)

**S2 Appendix. Questions and examples provided to LLM and Human scorers.** (A) List of questions for Cookie Theft picture description task. (B) Few-Shot Example 1. (C) Few-Shot Example 2.
(PDF)

## Acknowledgments

Special thanks to Katana Rader, Simone Camacho, and Andres Salcedo for assisting in the rating and consensus of the assessment responses. Additional thanks to James Galvin and the rest of the Comprehensive Center for Brain Health.

**Data sharing:** Data was obtained from the DementiaBank corpus of TalkBank [13], and is made available upon request following free membership admission. See their website at *dementia.talkbank.org* for more information about registering for membership, obtaining access to data, and other rules. Ratings from models, human raters, and consensus are available via publicly available repository [31].

## Author contributions

**Conceptualization:** Michael J. Kleiman.

**Data curation:** Michael J. Kleiman.

**Formal analysis:** Michael J. Kleiman.

**Funding acquisition:** Michael J. Kleiman.

**Investigation:** Michael J. Kleiman.

**Methodology:** Michael J. Kleiman.

**Validation:** Michael J. Kleiman.

**Visualization:** Michael J. Kleiman.

**Writing – original draft:** Michael J. Kleiman.

**Writing – review & editing:** Michael J. Kleiman.

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
