## [Decision Letter · Decision Letter 0]

9 Sep 2025

Response to Reviewers'. This file does not need to include responses to any formatting updates and technical items listed in the 'Journal Requirements' section below.'. This file does not need to include responses to any formatting updates and technical items listed in the 'Journal Requirements' section below.* A marked-up copy of your manuscript that highlights changes made to the original version. You should upload this as a separate file labeled 'Revised Manuscript with Track Changes'.'.* An unmarked version of your revised paper without tracked changes. You should upload this as a separate file labeled 'Manuscript'.'. If you would like to make changes to your financial disclosure, competing interests statement, or data availability statement, please make these updates within the submission form at the time of resubmission. Guidelines for resubmitting your figure files are available below the reviewer comments at the end of this letter. We look forward to receiving your revised manuscript. Kind regards, Thomas Kyumwa Kisimbi, RN, MA, MPA, MBAAcademic EditorPLOS Digital Health Thomas KisimbiAcademic EditorPLOS Digital Health Leo Anthony CeliEditor-in-ChiefPLOS Digital Healthorcid.org/0000-0001-6712-6626 **Journal Requirements:**

1. In the online submission form you indicate that your data is not available for proprietary reasons and have provided a contact point for accessing this data. Please note that your current contact point is a co-author on this manuscript. According to our Data Policy, the contact point must not be an author on the manuscript and must be an institutional contact, ideally not an individual. Please revise your data statement to a non-author institutional point of contact, such as a data access or ethics committee, and send this to us via return email. Please also include contact information for the third party organization, and please include the full citation of where the data can be found.

2. Please upload your main article file as a .doc, .docx or .rtf file.

4. Please provide separate main figure files in .tif or .eps format only and ensure that all files are under our size limit of 10MB.

For more information about how to convert your figure files please see our guidelines: https://journals.plos.org/digitalhealth/s/figures

5. Please include a separate legend or caption for Figure 1 in your manuscript.

6. We have noticed that you have uploaded Supporting Information files, but you have not included a list of legends. Please add a full list of legends for your Supporting Information files before or after the references list.

**Additional Editor Comments (if provided):** This manuscript introduces an innovative application of LLMs to a classic clinical task with strong relevance for digital health. It meets PLOS Digital Health’s publication criteria, but needs revisions to:

• Improve reproducibility (share prompts, code, sample outputs)

• Expand on equity and generalizability, including risks related to dialect, demographic variation, and AI bias

• Clarify clinical pathway - how would this tool be used? Assistive, supplementary, or diagnostic?

None of the concerns raised, by other reviewers or my own (see suggestions below) are fundamental flaws. They are issues of depth, clarity, and transparency, which are all addressable in a single round of major revision.

- Originality: Reviewers acknowledged this as a novel and timely use of LLMs in a clinically important task = Agreed. Original in applying prompt-engineered LLMs to a classic neuropsychological task.

- Statistical Rigor: Generally seen as sound, but some reviewers flagged missing benchmarks, lack of error analysis, or overreliance on MAE. Also of note, small sample size (n=25) limits generalizability= Agreed. Suggest expanding on fairness and error transparency, understood that this is proof-of-concept.

- Methodological Detail: Some reviewers felt prompt engineering methods were clearly described; others wanted more detail (e.g., why certain models were excluded). One reviewer suggested including newer models = I found the prompt engineering thoughtful and well-presented; new models as of publication incldued, wider range might be useful. Noted the lack of model/code sharing and dataset scale as limitations.

- Generalizability & Bias: At least one reviewer raised concerns about cultural/generalizability or representation in scoring models = I recommend addressing bias/fairness, especially dialect and cultural variation.

- Data & Code Availability: Several reviewers did not confirm open data/code, and this was seen as a possible gap = The manuscript would be stronger with prompt templates and outputs shared publicly.

**Reviewers' Comments:** Reviewer's Responses to Questions

**Comments to the Author**

1. Does this manuscript meet PLOS Digital Health’s publication criteria? Is the manuscript technically sound, and do the data support the conclusions? The manuscript must describe methodologically and ethically rigorous research with conclusions that are appropriately drawn based on the data presented.? Is the manuscript technically sound, and do the data support the conclusions? The manuscript must describe methodologically and ethically rigorous research with conclusions that are appropriately drawn based on the data presented.

Reviewer #1: Yes

Reviewer #2: Yes

Reviewer #3: Partly

Reviewer #4: Yes

Reviewer #5: Yes

2. Has the statistical analysis been performed appropriately and rigorously?

Reviewer #1: Yes

Reviewer #2: Yes

Reviewer #3: N/A

Reviewer #4: Yes

Reviewer #5: Yes

3. Have the authors made all data underlying the findings in their manuscript fully available (please refer to the Data Availability Statement at the start of the manuscript PDF file)?

The PLOS Data policy requires authors to make all data underlying the findings described in their manuscript fully available without restriction, with rare exception. The data should be provided as part of the manuscript or its supporting information, or deposited to a public repository. For example, in addition to summary statistics, the data points behind means, medians and variance measures should be available. If there are restrictions on publicly sharing data—e.g. participant privacy or use of data from a third party—those must be specified.requires authors to make all data underlying the findings described in their manuscript fully available without restriction, with rare exception. The data should be provided as part of the manuscript or its supporting information, or deposited to a public repository. For example, in addition to summary statistics, the data points behind means, medians and variance measures should be available. If there are restrictions on publicly sharing data—e.g. participant privacy or use of data from a third party—those must be specified.

Reviewer #1: Yes

Reviewer #2: Yes

Reviewer #3: Yes

Reviewer #4: No

Reviewer #5: Yes

4. Is the manuscript presented in an intelligible fashion and written in standard English?

Reviewer #1: Yes

Reviewer #2: Yes

Reviewer #3: Yes

Reviewer #4: Yes

Reviewer #5: Yes

Reviewer #1: This is an interesting paper with practical potential for neuropsychological assessment. You mention that the Cookie Theft task is comprised of 22 questions. It would be good to reference somewhere in your paper what type of question your LLM is trying to identify. This will enable the reader to gain more insight into how your LLM interprets the observations undertaken by the subject.

Reviewer #2: 1. While Claude 3 Opus performed best, the high token cost of this model raises questions about practical deployment. It would benefit the reader to include a concise cost-performance recommendation

2. Although consistency is highlighted, the manuscript could further discuss how explainability could affect clinician trust or integration into workflows.

3. There is limited discussion of how LLMs might perform across linguistic or demographic diversity. Given the clinical implications, a reflection on possible model biases or generalizability concerns is warranted.

4. The results would be strengthened by a figure or table summarizing MAE, ICC, and cost for all models and prompting strategies in a single view for easier comparison.

5. While briefly mentioned, further emphasis on how models like DeepSeek or reasoning-optimized LLMs could advance this field would enhance the discussion of scalability and technical evolution.

6. Consider explaining why JSON formatting was required earlier in the methods for readers unfamiliar with model deployment.

7. Clarify if models were zero-shot chain-of-thought only, or if basic zero-shot was also evaluated.

8. Consider including the mean and standard deviation for each model's MAE directly in the main results section (in addition to the figures/tables).

Reviewer #3: The study is timely and well-structured but has key limitations. Below are specific concerns that should be addressed to improve the rigor, clarity, and balance of the manuscript:

The sample size (n=25) is limited and may not support generalizable conclusions; further justification or discussion of this limitation is necessary.

The study does not explore or benchmark any medical-domain LLMs (such as, Med-PaLM 2, BioGPT), which may be better suited for clinical language understanding. This limits the scope and clinical relevance of the findings and should be acknowledged or addressed.

Inclusion of an untrained rater in the consensus scoring process may compromise the validity of the reference standard; the methodology and rationale should be clarified.

Multiple comparisons (110 ANOVAs) were conducted without indication of statistical correction, increasing the risk of false-positive findings.

Appendix B, which appears central to the scoring framework, is not included or summarized in the main text; representative items should be made accessible.

Conclusions regarding LLM “accuracy and reliability” appear overstated given the dataset scope and study constraints; more cautious interpretation is recommended.

The manuscript does not address potential real-world or ethical challenges, such as variability in language use, bias, or clinical safety implications.

Key terms such as MAE and CU are introduced without definition, which may limit accessibility to non-specialist readers.

Informal language and inconsistent reporting of model versions reduce clarity and professionalism; revisions for consistency and tone are advised.

Figures and tables lack fully informative captions and should be revised to be interpretable independently of the main text.

The discussion of prior literature is brief and lacks detailed methodological comparison; more thorough contextualization of findings is warranted.

Reviewer #4: The paper evaluates the effectiveness of five large language models (LLMs) in automatically scoring the Cookie Theft picture description task. I recommended that the authors consider the bellow suggestions:

1. The study uses a relatively small subset of 25 participants from the DementiaBank corpus, which may not sufficiently represent the diversity of cognitive impairments or demographic variability. This limits the generalizability of the findings and may not capture the full spectrum of language variations found in clinical populations.

2. The evaluation primarily focuses on mean absolute error (MAE) and interrater reliability, omitting other relevant metrics such as precision, recall, or agreement on specific content units. A broader set of evaluation measures could provide a more nuanced understanding of model behavior and limitations.

3. While few-shot learning, prompt chaining, and self-consistency are tested, the rationale for choosing these specific strategies over others is not fully explained. This weakens the methodological grounding and reduces reproducibility.

4. The paper does not include a detailed error analysis to show where or why models fail, especially in comparison to human raters. Understanding common misjudgments or failure cases would strengthen the validity of conclusions about model reliability.

5. The study evaluates only five LLMs, but does not compare their performance against simpler NLP-based scoring tools or statistical baselines. Including such comparisons would clarify the specific value added by advanced LLMs.

6. The manuscript lacks discussion on the ethical, clinical, or practical implications of using automated scoring in neuropsychological assessments, particularly regarding patient privacy, accountability, or potential misuse in clinical decision-making.

7. Details about prompts, model versions, and evaluation pipelines are summarized but not fully disclosed. Sharing full prompts and configuration settings is essential for reproducibility, especially when evaluating proprietary models like GPT-4o and Claude 3 Opus.

Reviewer #5: General Comments

This manuscript investigates the feasibility and reliability of using large language models (LLMs) to automate scoring of the Cookie Theft picture description task, a widely used neuropsychological tool. While the manuscript makes a strong and original contribution, several methodological and demographic reporting issues should be addressed before publication. In particular, more robust justification of the scoring reference standard, detailed statistical reporting, and expanded cost/efficiency analysis are needed to enhance the study’s impact and reproducibility.

Specific comments

1. The analysis was based on a subset of 25 participants selected from the DementiaBank corpus. Author stated that in line76-77: “The reduced size of the subset was necessary due to the requirement that each recording be manually transcribed, processed, and scored.” Please provide the explanation of the “requirement” with reference. Why only 25 with 13 cognitively normal and 12 cognitively impaired? How those 25 were selected from the 237 total sample?

2. The interrater reliability among human raters (ICC = 0.347) was notably low, which raises concerns about the validity of the scoring reference. Some justification or sensitivity analysis may be needed. Although the MAE and ICC are reported, author may consider reporting the effect sizes (e.g., Cohen’s d) alongside p-values in all inferential statistics with such small sample size. Also, a power analysis to support the adequacy of the sample size is needed.

3. There is minimal discussion of participant demographics beyond age, gender, and MMSE scores. For example, the language variation and education can influence performance on picture description tasks and LLM interpretability. A brief discussion of these covariates and their potential impact on the results are needed.

4. Line 89:” Each content unit was also formed into a simple True/False question and used by both the human raters as well as the LLM models to provide scores for the presence or absence of each content unit.” Please include a table for the list of the content unit questions used in the data coding process. In addition, the item-level error analysis, such as which content units caused most discrepancy could be included in the data analysis.

5. Although the paper discusses cost differences between models (e.g., Claude Opus vs GPT-4o), it lacks quantitative detail. Reporting actual processing time, token counts, and estimated dollar costs per transcript would provide clearer guidance for real-world deployment.

6. Consider clarifying the low ICC for GPT-4o reported in Table 2 (0.110). This appears inconsistent with other reported findings and may need to be checked or contextualized.

7. Including a table with examples of scoring errors (model vs. human) would help illustrate the types of disagreements observed.

8. The discussion could benefit from a brief reflection on the ethical implications of using LLMs in cognitive assessment (e.g., fairness, bias, transparency).

9. Reference formatting is inconsistent in a few places; ensure alignment with journal style guidelines. Table formatting and figure references (e.g. Figure 1) should align with journal style guidelines as well.

**Do you want your identity to be public for this peer review?** For information about this choice, including consent withdrawal, please see our Privacy Policy..

Reviewer #1: No

Reviewer #2: No

Reviewer #3: No

Reviewer #4: No

Reviewer #5: No

**Figure resubmission:** While revising your submission, please upload your figure files to the Preflight Analysis and Conversion Engine (PACE) digital diagnostic tool, https://pacev2.apexcovantage.com/. PACE helps ensure that figures meet PLOS requirements. To use PACE, you must first register as a user. Registration is free. Then, login and navigate to the UPLOAD tab, where you will find detailed instructions on how to use the tool. If you encounter any issues or have any questions when using PACE, please email PLOS at figures@plos.org. Please note that Supporting Information files do not need this step. If there are other versions of figure files still present in your submission file inventory at resubmission, please replace them with the PACE-processed versions. **Reproducibility:** To enhance the reproducibility of your results, we recommend that authors of applicable studies deposit laboratory protocols in protocols.io, where a protocol can be assigned its own identifier (DOI) such that it can be cited independently in the future. Additionally, PLOS ONE offers an option to publish peer-reviewed clinical study protocols. Read more information on sharing protocols at https://plos.org/protocols?utm_medium=editorial-email&utm_source=authorletters&utm_campaign=protocols To enhance the reproducibility of your results, we recommend that authors of applicable studies deposit laboratory protocols in protocols.io, where a protocol can be assigned its own identifier (DOI) such that it can be cited independently in the future. Additionally, PLOS ONE offers an option to publish peer-reviewed clinical study protocols. Read more information on sharing protocols at https://plos.org/protocols?utm_medium=editorial-email&utm_source=authorletters&utm_campaign=protocols

---

## [Decision Letter · Decision Letter 1]

8 Apr 2026

Evaluating Large Language Model Performance and Reliability in Scoring Picture Description Tasks for Neuropsychological Assessment

PDIG-D-25-00277R1

Dear Dr. Kleiman,

We are pleased to inform you that your manuscript 'Evaluating Large Language Model Performance and Reliability in Scoring Picture Description Tasks for Neuropsychological Assessment' has been provisionally accepted for publication in PLOS Digital Health.

Best regards,

Henry Horng-Shing Lu

Section Editor

PLOS Digital Health

**Additional Editor Comments (if provided):**

**Reviewer Comments (if any, and for reference):**

Reviewer's Responses to Questions

**Comments to the Author**

Reviewer #1: All comments have been addressed

publication criteria? Is the manuscript technically sound, and do the data support the conclusions? The manuscript must describe methodologically and ethically rigorous research with conclusions that are appropriately drawn based on the data presented.? Is the manuscript technically sound, and do the data support the conclusions? The manuscript must describe methodologically and ethically rigorous research with conclusions that are appropriately drawn based on the data presented.

Reviewer #1: Yes

3. Has the statistical analysis been performed appropriately and rigorously?

Reviewer #1: Yes

4. Have the authors made all data underlying the findings in their manuscript fully available (please refer to the Data Availability Statement at the start of the manuscript PDF file)?

The PLOS Data policy requires authors to make all data underlying the findings described in their manuscript fully available without restriction, with rare exception. The data should be provided as part of the manuscript or its supporting information, or deposited to a public repository. For example, in addition to summary statistics, the data points behind means, medians and variance measures should be available. If there are restrictions on publicly sharing data—e.g. participant privacy or use of data from a third party—those must be specified.requires authors to make all data underlying the findings described in their manuscript fully available without restriction, with rare exception. The data should be provided as part of the manuscript or its supporting information, or deposited to a public repository. For example, in addition to summary statistics, the data points behind means, medians and variance measures should be available. If there are restrictions on publicly sharing data—e.g. participant privacy or use of data from a third party—those must be specified.

Reviewer #1: Yes

5. Is the manuscript presented in an intelligible fashion and written in standard English?

Reviewer #1: Yes

Reviewer #1: Thank you for this interesting paper and research. I have reviewed the responses provided and the author appears to have addressed the points raised. I am satisfied with these responses.

**Do you want your identity to be public for this peer review?** For information about this choice, including consent withdrawal, please see our Privacy Policy..

Reviewer #1: No
